# A Past Genetic Bottleneck from Argentine Beans and a Selective Sweep Led to the Race Chile of the Common Bean (*Phaseolus vulgaris* L.)

**DOI:** 10.3390/ijms25074081

**Published:** 2024-04-06

**Authors:** Osvin Arriagada, Bárbara Arévalo, Igor Pacheco, Andrés R. Schwember, Lee A. Meisel, Herman Silva, Katherine Márquez, Andrea Plaza, Ricardo Pérez-Diáz, José Pico-Mendoza, Ricardo A. Cabeza, Gerardo Tapia, Camila Fuentes, Yohaily Rodríguez-Alvarez, Basilio Carrasco

**Affiliations:** 1Centro de Estudios en Alimentos Procesados (CEAP), Av. Lircay s/n, Talca 3480094, Chile; barevalo@ceap.cl (B.A.); kmarquez@ceap.cl (K.M.); aplaza@ceap.cl (A.P.); jrperezd@gmail.com (R.P.-D.); cmla.fuentes@gmail.com (C.F.); 2Instituto de Nutrición y Tecnología de los Alimentos (INTA), El Líbano 5524, Santiago 7830490, Chile; igor.pacheco@inta.uchile.cl (I.P.); lmeisel@inta.uchile.cl (L.A.M.); 3Departamento de Ciencias Vegetales, Facultad de Agronomía y Sistemas Naturales, Pontificia Universidad Católica de Chile, Santiago 7820436, Chile; aschwember@uc.cl (A.R.S.); yrodriguez4@uc.cl (Y.R.-A.); 4Laboratorio de Genómica Funcional & Bioinformática, Departamento de Producción Agrícola, Facultad de Ciencias Agronómicas, Universidad de Chile, Av. Santa Rosa 11315, Santiago 8820808, Chile; hesilva@uchile.cl; 5Facultad de Ingeniería Agronómica, Universidad Técnica de Manabí, Portoviejo 130105, Ecuador; newthon.pico@utm.edu.ec; 6Laboratorio de Nutrición Vegetal, Departamento de Producción Agrícola, Facultad de Ciencias Agrarias, Universidad de Talca, Talca 3460000, Chile; rcabeza@utalca.cl; 7Unidad de Recursos Genéticos, Instituto de Investigaciones Agropecuarias (INIA Quilamapu), Chillán 3800062, Chile; gtapia@inia.cl; 8Programa de Doctorado en Ciencias Agrarias, Facultad de Ciencias Agrarias, Universidad de Talca, Talca 3460000, Chile

**Keywords:** Andean pool, common bean, genetic diversity, genetic structure, race Chile

## Abstract

The domestication process of the common bean gave rise to six different races which come from the two ancestral genetic pools, the Mesoamerican (Durango, Jalisco, and Mesoamerica races) and the Andean (New Granada, Peru, and Chile races). In this study, a collection of 281 common bean landraces from Chile was analyzed using a 12K-SNP microarray. Additionally, 401 accessions representing the rest of the five common bean races were analyzed. A total of 2543 SNPs allowed us to differentiate a genetic group of 165 accessions that corresponds to the race Chile, 90 of which were classified as pure accessions, such as the bean types ‘Tórtola’, ‘Sapito’, ‘Coscorrón’, and ‘Frutilla’. Our genetic analysis indicates that the race Chile has a close relationship with accessions from Argentina, suggesting that nomadic ancestral peoples introduced the bean seed to Chile. Previous archaeological and genetic studies support this hypothesis. Additionally, the low genetic diversity (π = 0.053; uHe = 0.53) and the negative value of Tajima’ D (D = −1.371) indicate that the race Chile suffered a bottleneck and a selective sweep after its introduction, supporting the hypothesis that a small group of Argentine bean genotypes led to the race Chile. A total of 235 genes were identified within haplotype blocks detected exclusively in the race Chile, most of them involved in signal transduction, supporting the hypothesis that intracellular signaling pathways play a fundamental role in the adaptation of organisms to changes in the environment. To date, our findings are the most complete investigation associated with the origin of the race Chile of common bean.

## 1. Introduction

The race Chile of common bean has attracted significant interest due to a recent study indicating that modern Chilean landraces maintain levels of variability and genetic identity like the ancestral common bean found in archaeological sites in northern Argentina dated between 2500 and 600 years before the present (BP), suggesting that the race Chile is a genetic reservoir of the current Andean gene pool [1]. Additionally, Bellucci et al. [2] reported that the Andean gene pool was the first to be introduced to Europe after Francisco Pizarro’s expedition to northern Peru in 1529, in which most of the Andean genetic background of the European common bean was introgressed by the races Nueva Granada and Chile. These results are not surprising since archaeological studies show that the common bean has been cultivated ancestrally from the north [3,4] to the south [5,6] of Chile since at least 3000 years BP.

Two gene pools, Mesoamerican and Andean, diverged from a common ancestor ~165,000 years ago, and are morphologically and genetically different [7]. As a result of this divergence, six races of common bean emerged that differ in ecological adaptation, geographic range, morpho-agronomic traits, and phaseolin types [8]. The Durango, Jalisco, and Mesoamerica races originated from the Mesoamerica gene pool, while the New Granada, Peru, and Chile races belong to the Andean genetic pool [8]. Although the race Chile belongs to the Andean pool, its origin is still unknown. The domestication of the Chilean common bean was the most intense in the Americas, based on the fact that it is the only country where it does not exist in the wild form [9]; and ancestrally, its consumption has been exclusively associated with indigenous communities, where an intense selection process may have occurred. This situation would have modified its genetic structure and ability to adapt to the different environmental conditions present in Chile. In fact, the Chilean common bean is cultivated from Arica (18°28′40″ S–70°19′05″ W, average temperature of 22 °C, and precipitation of 0–1 mm per year) to Chiloé (42°40′36″ S–73°59′36″ W; average temperature of 9.8 °C, and precipitation of ~1200 mm per year), which have allowed it to possess adaptations and characteristics that are unique, differing from the other common bean races described around the world [9]. From a nutritional point of view, Chilean landraces of common bean have shown a wide variability for seed macro- and micronutrients such as N, Fe, and Zn [10], a high content of serine, a wide variability of total phenolic content, and a range of healthy proteins, suggesting that this genetic resource is a valuable source of plant-based protein for direct consumption and/or for the development of functional ingredients [11]. In this context, the race Chile of common bean represents a valuable genetic resource with unique local adaptations that can be key to coping with the challenges of global climate change and a nutritionally valuable crop necessary to ensure food safety. However, little attention has been paid to the characterization of its genetic diversity, population structure, linkage disequilibrium (LD), and domestication process.

Studies have detected low to moderate levels of genetic diversity within the race Chile genotypes using biochemical and DNA markers such as phaseolin seed protein, isozyme [12], SSR [13], and SNP markers [1,2]. However, these results should be considered cautiously due to the limited number of accessions evaluated. The most significant study regarding Chilean common bean genetic diversity was performed by Becerra et al. [9], who analyzed 237 Chilean landraces from the INIA (National Agricultural Research Institute, Chillán, Chile) genebank. The results showed a moderate level of genetic diversity within these accessions, which was closely related to race Nueva Granada according to the Euclidean distance of 0.12.

The development of microarrays of SNPs for common bean [14], together with the release of the reference genome [7], are powerful molecular tools for carrying out depth genetic characterization. Therefore, the main objective of this investigation was to perform a complete genetic and genomic characterization of the Chilean common bean. This information will be crucial to identify and determine the origin of the race Chile of common bean, for conserving and valorizing this unique genetic resource, and for developing strategies for genetic improvement in this species, considering the ongoing effects of climate change and the demand for more nutritious crops.

## 2. Results

### 2.1. SNP Genotyping

Based on the Phytozome database (https://phytozome-next.jgi.doe.gov/ accessed on 23 October 2023), a total of 3013 SNP markers from the BARCBean12K_3 Infinium SNP array, distributed across the 11 chromosomes, were aligned on the genome of common bean v2.1 (Appendix A). The final marker set comprised 2543 SNPs, primarily concentrated in the regions close to the telomeres. The number of markers varied from 139 to 292 on chromosomes Pv06 and Pv05, respectively, with an average of 231 SNPs per chromosome. Physical chromosome length ranged from 30.99 to 62.95 Mpb on chromosomes Pv06 and Pv08, respectively. On average, a marker density of one SNP every ~0.20 Mpb and a SNP gap of ~0.40 Mpb were estimated. The highest density was found in the region between 38 and 40 Mpb on chromosome Pv05, while the largest SNP gap (3.5 Mpb) was found at ~14.7 Mpb on chromosome Pv10 (Appendix A).

### 2.2. Population Structure and Relationships among Individuals

After quality filters, a total of 667 accessions were considered for subsequent analyses. Population structure analysis indicated K = 2 as the most probable number of genetic groups, which is in accordance with the two existing common bean gene pools (Appendix A). One hundred and sixty-five SNPs (Fst ≥ 0.9) enabled the differentiation between these two gene pools (Appendix A). Based on the membership coefficient (Q_i_ > 0.6) in K = 2, 99.1% of the accessions were assigned to a specific group, and only six were categorized as an admixture (Figure 1A, Appendix A). One group of 311 accessions had a predominantly Andean origin, while the second group with 350 accessions had a predominantly Mesoamerican origin. Fifty-nine accessions (fifty from Chile and nine from Peru) that were initially classified to the Andean gene pool according to passport information were assigned to the Mesoamerican group. Conversely, 46 accessions from the Mesoamerican gene pool (mainly from Mexico and Colombia) were grouped within the Andean genetic group (Appendix A). The principal component analysis (PCA) also showed a division between the Mesoamerican and the Andean gene pools (Figure 1B). A new PCA was performed to investigate the subdivisions of the accessions according to the country of origin using passport information. In this sense, the first principal component (PC1) differentiated accessions from the Mesoamerica and South America regions, and the second principal component (PC2) differentiated the accessions from Mexico and Central America within the Mesoamerican group, and the accessions from Chile, Peru, and Brazil in the Andean gene pool. Accessions classified as ‘Others’ mostly corresponded to advanced breeding lines that showed intermediate values between both principal components (Figure 1C).

To further investigate the substructures shown in Figure 1B, the accessions from the Mesoamerican and Andean groups according to structure result were analyzed independently. In this sense, the 350 accessions belonging to the Mesoamerican group were divided into two subgroups (K = 2), hereafter indicated as Mesoamerican 1 (M1) and Mesoamerican 2 (M2) (Appendix A). A total of 235 accessions were grouped into the M1 group, which was mainly constituted by accessions from Mexico (142). The 103 accessions that were grouped into the M2 group are mainly from Mexico (31) and Central America [Costa Rica (13), El Salvador (8), and Nicaragua (9)], while the remaining 12 accessions belonging to the Mesoamerican group were classified as an admixture. On the other hand, the 311 accessions belonging to the Andean group were divided into three subgroups (K = 3; Appendix A), hereafter called Andean 1 (A1), Andean 2 (A2), and Andean 3 (A3). Group A1 consisted of 10 accessions from Chile (7), Colombia (2), and Guatemala (1). A total of 66.1% of the accessions from Peru were grouped into the A2 group. Interestingly, 187 accessions (Q_i_ > 0.6) from the Chilean germplasm collection were grouped exclusively into the A3 group, indicating that these accessions correspond to the race Chile of common bean. After removing duplicated genotypes, 165 accessions of the race Chile were retained, within which 90 were classified as pure accessions (Q_i_ > 0.99), highlighting the types ‘Tórtola’, ‘Coscorrón’, ‘Frutilla’, ‘Sapito’, and ’Burros’. The genetic relationships between those pure accessions of race Chile are shown in Appendix A, and the seed shape and color are shown in Appendix A.

### 2.3. Genetic Diversity According to Population Structure

Regarding the 2543 polymorphic SNPs, 94.93% and 88.64% were polymorphic within the Mesoamerican and Andean groups, respectively. Within the Andean genetic pool, the A3 subgroup (Chile) was the one that presented the highest percentage of polymorphic loci (87.06%). While in the Mesoamerican genetic pool, the M1 subgroup had 93.87% of polymorphic loci. In general, according to parameters of genetic diversity, such as unbiased expected heterozygosity (uHe), Shannon’s information index (I), and nucleotide diversity (π), the accessions from the Mesoamerican (uHe = 0.300; I = 0.446; π = 0.300) gene pool have a higher level of genetic diversity compared to accessions from the Andean (uHe = 0.099; I = 0.177; π = 0.101) gene pool (Table 1). Genome-wide Tajima’s D value was positive and significant for the Mesoamerican gene pool (D = 3.203), whereas the Andean gene pool showed a negative value (D = −0.962). Based on the subgroups, the subgroup M1, corresponding to accessions coming mainly from Mexico, has the highest genetic diversity (uHe = 0.287, I = 0.431, π = 0.286) in contrast to the M2 group, whose accessions are mainly from Central America (uHe = 0.177; I = 0.273; π = 0.176). Tajima’s D values were positive in both groups; however, it was significant only in the M1 group. Within the subgroups belonging to the Andean gene pool, the genetic subgroup race Chile (A3) was the one that showed the lowest level of genetic diversity (uHe = 0.053; I = 0.096; π = 0.053), followed by the A2/race Peru subgroup (uHe = 1.102, I = 1.165, and π = 0.102). In general, the race Chile showed the largest deviation from neutrality (D = −1.371). Interestingly, the A1 subgroup had similar levels of genetic diversity relative to the M2 subgroup and is the only one with positive values of Tajima’s D (D = 0.370) within the Andean subgroups (Table 1). High (Fst = 0.305) to very high (Fst = 0.841) levels of genetic differentiation were found among the subgroups M1 and M2, and M2 and A3, respectively (Appendix A).

Given that our research interest was to characterize the race Chile, the nucleotide diversity and Tajima’s D were estimated through the genome according to the gene pools [Andean (A1 + A2) and Mesoamerican (M1 + M2)] and subgroup race Chile (Figure 2). For the accessions belonging to the race Chile, negative Tajima’s D values were observed on all the chromosomes, except for some regions of chromosomes Pv02 and Pv05, which coincide with the regions that present higher levels of nucleotide diversity. These results indicate a balanced selection in these genomic regions within the race Chile. To identify regions that had significant Tajima’s D values, an analysis by marker pairs was performed (Appendix A), where regions on chromosomes Pv05 (position 36,195,096–36,365,141 bp) and Pv07 (2,469,407–2,469,664 bp) were significant (D value > 2). For the Andean group, Tajima’s D values varied widely across the genome with primarily negative values, except on chromosomes Pv04 and Pv09. In general, the nucleotide diversity was higher than that observed in the race Chile, except on regions of chromosomes Pv02 and Pv05. For the Mesoamerican group, the nucleotide diversity was higher on all chromosomes compared to the Andean group and the race Chile. Additionally, Tajima’s D values were positive and widely variable across the whole genome.

Finally, the genetic relationships between some pure race Chile and accessions from the Andean gene pool are shown in Figure 3, indicating that the race Chile has a close genetic relationship with accessions from Argentina, and that both have a common origin from the Peru and Bolivia genotypes.

### 2.4. Linkage Disequilibrium and Haplotypes Blocks

The genomic kinship and population structure matrix obtained for the Mesoamerican, Andean, and race Chile groups using 2543 SNP were used to correct the LD. In general, LD decay more rapidly on all chromosomes of the Mesoamerican group compared to the Andean and Chile groups (Figure 4). On chromosomes Pv02 and Pv10, the LD decay pattern is similar. In the Mesoamerican group, the LD decays from values of r^2^_vs_ ~0.4 to values less than r^2^_vs_ ~0.1 at distances lower than 5 Mpb. In the race Chile of common bean, the LD decays slowly on all chromosomes except on Pv08 and Pv10, where LD decayed faster than in the Andean group. The chromosomes Pv03, Pv06, and Pv11 show the slowest LD decay pattern, where they decay to values close to ~0.2 at distances of approximately ~30 Mpb.

A total of 333 haplotype blocks were identified across the common bean genome in the three genetic groups analyzed. At the chromosome level, the number of haplotype blocks varied from two to sixty-nine on chromosomes Pv03 and Pv02, respectively. More haplotype blocks were found in the Mesoamerican group (172 blocks) compared to the Andean group (110 blocks). In the Mesoamerican group, unique blocks varied from zero on chromosome Pv03 to 19 on chromosome Pv08. In contrast, in the Andean group, the unique blocks varied from zero on chromosomes Pv06 and Pv08 to four on chromosome Pv04. The size of the haplotype blocks was greater in the race Chile (an average of 202.8 Kb per block) than in the Andean (141.6 kb) and the Mesoamerican (60.3 Kb) groups, which agrees with the levels of LD decay for each group. For the race Chile, 51 haplotype blocks were identified on all chromosomes, except on Pv03, Pv04, and Pv06. The chromosomes Pv09 and Pv11 had the fewest number of blocks (1), whereas Pv02 had the highest number of blocks (16). Interestingly, twelve unique blocks were identified on the race Chile genome, varying from one on chromosomes Pv09 and Pv11 to five on Pv10 (Table 2).

### 2.5. Candidate Gene Identification and GO Gene Enrichment

A total of 235 genes were identified within the haplotype blocks found for the race Chile. After the GO analysis, we identified 235 sequences associated with 678 GO categories. Within the three main categories, the molecular function (MF) category was the most abundant (261 annotations divided into 47 GO subcategories), followed by the biological process (BP) (222 annotations, divided into 29 GO subcategories) and cellular component (CC) categories (195 annotations, divided into 20 GO subcategories) (Figure 5A, Appendix A). Genes that encode binding proteins were the most represented within the MF category. Genes associated with oxidoreductase activity were also highly represented. In the case of BP, genes associated with transduction signals and biosynthetic processes were the most abundant in that category. In the CC category, membrane corresponds to the most abundant term, followed by nucleus and cytoplasm (Figure 5B). Finally, 11 genes within the region on chromosome Pv05 with a significant Tajima value were found (Table 3). In contrast, no genes were found within the region on chromosome Pv07.

## 3. Discussion

### 3.1. Archaeological Records of Common Bean in Chile

Two independent domestication events in Mexico and South America about 8000 years ago gave rise to two morphologically and genetically differentiated groups known as Mesoamerican and Andean gene pools [7]. Mexico was proposed to be the center from which the common bean originated and from where it spread across South America [15]. Specifically, the Oaxaca Valley was proposed as the most likely origin of domestication in Mesoamerica [16,17]. In the Andes, the regions of northern Argentina and southern Bolivia have been proposed as the putative center of domestication [16,17].

In Chile, two explorations performed in the north and central zone of the country confirmed the absence of wild beans [9], discarding the hypothesis of an independent domestication process that led to the race Chile [13]. On the contrary, antecedents and our results indicate that the race Chile of common bean is a direct descendant of the seeds from the primary center of origin in Northwestern Argentina, whose entry into Chile was probably through the Andes Mountain range into temperate latitudes of the northern and the central part of Chile before the Inca conquest [4,18]. This hypothesis is supported by the fact that modern Chilean beans show a close genetic relationship with Argentine wild accessions [13] and with the ancient seeds of northern Argentina dated between 2500 and 600 years BP [1]. This event is consistent with the ancient common bean seeds found in the archaeological sites of the Atacama Desert and San Pedro Viejo de Pichasca in the north of Chile, dated between 3000 and 1450 years BP [3,4]. In southern Chile, ancient seeds were found in Isla Mocha and Cueva de los Catalanes archaeological sites dated 1300–1000 years BP [5,6]. Non-isolation by distance between accessions from northern and southern Chile was determined by the Mantel test (*p* = 0.370), suggesting that local farmers maintained the traditional practice of seed exchange at that time.

Interestingly, a total of six accessions from northwest Argentina (Tucuman, Salta, and Jujuy) were evaluated in this study. These were classified as admixtured with an ancestry of ~50% of their genome from the race Chile. According to neighbor-joining analysis, the idea that the common bean race Chile originated from the entry into Chile of genotypes from Argentina is supported, and both genetic resources derive from genotypes originating from Peru. This result supports the hypothesis that *P. vulgaris* from northern Peru–Ecuador is a relict population representing a fraction of the genetic diversity of the ancestral population that migrated from Mexico [15]. However, we do not have information on the degree of domestication of all the accessions used in this study, which is essential since modern Argentine accessions are not closely related to ancient local accessions [1]. Therefore, more complete studies using wild, ancient, and modern seeds from the primary and secondary centers of domestication of the species are necessary to improve our understanding of the origin of the race Chile of common bean.

### 3.2. Population Structure and Genetic Diversity

The accessions of *P. vulgaris* evaluated in this study, which cover a large part of the geographic regions of both the primary and the secondary domestication centers, were consistently grouped into Mesoamerican and Andean gene pools. The Mesoamerican gene pool is characterized by greater levels of genetic diversity than the Andean gene pool [17], which is consistent with our results. This is probably due to a small founding population and a more substantial bottleneck effect that lasted ~76,000 years in the Andes before the domestication process [7]. After domestication, local adaptation resulted in different eco-geographical races of common bean within each genetic pool. In this sense, the accessions belonging to the Mesoamerican and the Andean gene pool generally are grouped into two or more genetic subgroups [2,13], which depends on the geographical origin of the accessions and the degree of domestication.

Two Mesoamerican (M1 and M2) genetic subgroups were identified, including genotypes from Chile, mainly in the M1 group. This result is not surprising considering that some accessions present in Chile are probably the result of genomic introgressions by artificial or natural crosses from the Mesoamerican gene pool [9,12]. Few studies have included accessions of the race Chile, and most of them have not been able to differentiate the race Chile from the races Nueva Granada and Peru. For example, the studies carried out by Blair et al. [13] and Blair et al. [19] analyzed six and eight race Chile accessions, respectively, with 33 SSR markers, showing differentiation between the races Nueva Granada and Peru, while the race Chile could not be distinguished from the others. In this study, the Andean gene pool was divided into three genetic subgroups according to allelic frequencies, in which A3 group consisted of 187 accessions exclusively from Chile. Recent studies using a small number of Chilean accessions, but many markers have shown that the race Chile differs genetically from the two other Andean races [2,9]. In those studies, the type accessions ‘Tórtola’, ‘Coscorrón’, ‘Burros’, and ‘Frutilla’ were used, which coincide with the accessions classified as pure in this study.

The genetic diversity of the race Chile has been quantified as low and closely related to the race Nueva Granada [9,12], probably a consequence of the self-pollinating reproduction of the common bean, the absence of wild accessions, and the possible origin of the race Chile. We characterized the nucleotide diversity and neutrality at the genomic level. We found low diversity and negative Tajima’s D values in almost all chromosomes, except in some regions of the chromosomes Pv02 and Pv05. This low nucleotide diversity indicates relatively low effective population size or past bottlenecks. In contrast, negative Tajima’s D values may indicate recent population expansion after a bottleneck or selective sweeps due to new advantageous mutations [20], which agrees with the origin of race Chile and its local adaptations. Trucchi et al. [1] indicated that the chromosomes Pv01, Pv02, and Pv10 are involved in the domestication process of the Andean genetic pool. Moreover, Papa et al. [21] showed that farmers and breeders have less exploited regions linked to the domesticated loci, and therefore, the ones where the highest diversity is located, which is in accordance with the level of genetic diversity found on chromosomes Pv02 and Pv05.

### 3.3. Local Adaptation

Common bean presents variation in seed size and shape, seed coat color and patterns, growth habit, and phenological traits [2]. The genetic group M1 showed the phaseolin type ‘S’ and presented seed sizes varying from small (<30 g/100 seed) to medium (35 g/100 seed). The M2 group showed an ‘S’ or ‘Sd’ phaseolin type and an average small seed size of 25 g/100 seed [22].

The race Chile has been mainly described through its morphology, where the plants have relatively small leaves, pink to white flowers, medium-sized pods that are not very fibrous, and round to kidney-shaped seeds with three to five seeds per pod [8,10,23]. The main agronomic traits of the race Chile are the predominantly type III growth habit (prostrate or semi-prostrate), intermediate to late maturity, and a seed size that varies from medium (25–40 g/100 seed) to large (>40 g/100 seed) [23]. However, it presents susceptibility to common yellow, cucumber, and alfalfa mosaic viruses [9]. From a nutritional standpoint, the race Chile has been poorly characterized. Paredes et al. [10] reported a wide variability for micronutrients such as Fe (68.9 to 152.4 mg/kg) and Zn (27.9 to 40.7 mg/kg). The protein content varied from 18 to 26 g/100 g [10], and phaseolin type ‘C’ and ‘T’ are the most abundant [12]. Recently, a more comprehensive study confirmed the wide variation for different nutritional components (soluble protein, amino acid, sugars, mineral composition, antioxidant capacity, and total phenolic content) and antinutritional (raffinose) components in race Chile [11]. They concluded that common bean types such as ‘Tórtola’, ‘Frutilla’, ‘Sapito’, and ‘Palo’, among others, have high nutritional value, confirming the value of this genetic resource.

Common bean plays an important economic, social, and nutritional role. The genotypes cultivated in Chile by farmers are mainly local landraces and a small percentage of improved varieties [9]. Common bean is cultivated in different agroclimatic conditions from the north of the country (Arica: 18°28′ S; 70°19′ W), which is characterized by a coastal desert climate with an average temperature of 18 °C and lacks rainfall (0 mm/year), to areas as remote as the Chiloé Island (42°40′ S; 73° 59′ W) in the south of Chile, characterized by a temperate maritime rainy climate with average temperatures of 11 °C and annual rainfall above 2000 mm [23]. This wide distribution of cultivation, together with the local adaptations, has allowed this species to acquire unique characteristics of agronomic and nutritional interest.

### 3.4. Identification of Candidate Genes and GO Analyses

The identification and characterization of the genes present in the haplotype blocks showed that genes related to transduction signals and biosynthetic processes were the most abundant in the category of biological processes. These results demonstrate the importance of these genes in the regulation and execution of a wide variety of biological processes [24]. The abundance of genes involved in signal transduction shows that they are essential for intracellular communication and coordination in response to environmental changes, supporting the hypothesis that intracellular signaling pathways play a fundamental role in the adaptation of organisms to changes in the environment. In Gene Ontology Graphs of Biological Processes (Appendix A), it is observed that the most specialized terms (child nodes) correspond to responses to external factors such as salt, fluoride, and oxygen. On the other hand, the abundance of genes related to biosynthetic processes highlights the importance of synthesizing molecules and compounds necessary for the proper growth and functioning of organisms. Overexpression or underexpression of biosynthetic genes may be a key mechanism in an organism’s response to environmental stress, allowing it to reallocate resources to produce compounds essential for survival.

Specifically, eleven genes were found within the region on chromosome Pv05 with a significant Tajima D value in the race Chile. Among these, two genes that code for an A20/AN1-like zinc finger family protein (*Phvul.005G125000*) and a serine-threonine protein kinase (*Phvul.005G124800*) have been associated with the tolerance to multiple abiotic and biotic stresses in crops [25,26]. In addition, the A20/AN1-like zinc finger family members are involved in plant growth regulation by affecting GA biosynthesis [27]. A gene encoding for vacuolar protein-sorting-associated protein 4 (VPS4; *Phvul.005G125100*), present in this region, has been mapped as bc-4 locus as a strong candidate in the resistance to the Bean Mosaic Common Virus (BMCV) [28], probably explaining the susceptibility of the race Chile to some mosaic plant viruses [9]. Interestingly, a gene (*Phvul.005G126300*) that codes for a transcription factor superfamily APETALA2/Ethylene Responsive Factor (AP2/ERF), which participates in processes such as tolerance to abiotic (salinity, drought, and temperature fluctuation) stress, biotic resistance, and plant development in common bean [29], was identified in the Chile race. *Phvul.005G126300* has been annotated as a member of DREB gene family, group A-1, whose members are inducible under certain stress conditions [30]. These genes may have played a key role in adaptation to the different climatic conditions present in Chile where the bean is distributed. Taking all together, these preliminary results provide a solid foundation for future research into how these specific genes, and others, influence and regulate the response of race Chile of common bean to environmental stress, which our research group is currently studying.

## 4. Materials and Methods

### 4.1. Plant Material

A collection of 682 common bean accessions was used. A total of 316 accessions were provided by the Instituto de Investigaciones Agropecuarias (Chile’s National Agricultural Research Institute, INIA-Quilamapu, Chillán, Chile), in which 281 accessions correspond to landraces from Chile. The remaining 366 accessions were extracted from Kuzay et al. [22], which come from USDA-ARS, the National Genetic Resources Program, and the Germplasm Resources Information Network [GRIN]. The complete list of common bean accessions and their passport information is provided in the Appendix A.

### 4.2. SNP Genotyping and Filtering

For DNA purification, total genomic DNA was extracted from young leaf using the DNeasy^®^ Plant Mini Kit (Qiagen, Venlo, The Netherlands) according to the manufacturer’s instructions. DNA concentration and quality were assessed with the Qubit Fluorometer using Qubit dsDNA HS Assay Kit (Thermofisher, Waltham, MA, USA) and agarose gel electrophoresis (0.8% *w*/*v*), respectively.

DNA samples were genotyped with the Illumina (Illumina Inc., San Diego, CA, USA) BARCBean12K_3 Infinium SNP array [14], which contains 11,225 SNP markers distributed across the 11 common bean chromosomes. SNP calling was conducted with the genotyping module V2011.1 of GenomeStudio software (Illumina Inc.) according to Cichy et al. [31]. To obtain the collection of 682 accessions analyzed, common SNPs were identified between the accessions published by Kuzay et al. [22] and the 316 accessions provided by the INIA-Quilamapu. Due to the difference in nomenclature in the SNPs compared, a translation was carried out, which allowed us to homologate the information to the nomenclature of the BACBean12K chip and identify the common SNPs.

Monomorphic SNP, SNPs with minor allele frequency (MAF) of less than 5%, and those SNPs and accessions containing 20% or more of missing values were removed using TASSEL 5.2 software [32]. Because the common bean is an autogamous species, the genotypes with 15% or more heterozygotes were also eliminated. After SNP filtering, the remaining missing data were imputed using the software Beagle 5.4 [33].

### 4.3. Population Structure Analysis

The population structure was inferred in Structure v2.3.4 [34]. An admixture ancestry model with correlated allele frequencies and no prior information of population origin was used. The putative number of subpopulations (*K*) ranged from 1 to 7. For each *K*, ten replicates were performed with a burn-in period of 100,000 steps followed by 1,000,000 MCMC iterations. The optimal *K* value was determined in Structure Harvester (https://taylor0.biology.ucla.edu/structureHarvester/ accessed on 5 September 2023) using the ad hoc statistic Δ*K* [35]. Each accession was assigned to a genetic group based on the inferred ancestry, which was classified as pure accession (Q_i_ ≥ 0.99), accessions belonging to a genetic group with a slight level of admixture (0.6 > Q_i_ < 0.99), or admixed accessions (Q_i_ ≤ 0.6) [2]. Subsequently, the population structure of the accessions was evaluated by principal component analysis implemented in ade4 [36] and adegenet [37] and visualized with ggplot2 [38] packages. To identify genetic subgroups within each gene pool, a new Structure run was performed independently for each genetic group using the same parameters described above but varying *K* from 1 to 5. Finally, a dendrogram was constructed with some Chilean accessions classified as pure to show the genetic relationship with representatives of the Andean and the Mesoamerican genetic pools. It was implemented using the function aboot implemented in the poppr package [39] based on Nei’s genetic distance with 1000 bootstrap replicates. The resulting tree was visualized with FigTree v1.4 (http://tree.bio.ed.ac.uk/software/figtree/ accessed on 15 September 2023).

### 4.4. Genetic Diversity by Population and Along the Genome

The descriptive diversity statistics such as the mean number of alleles (Na), the mean effective number of alleles (Ne), Shannon diversity index (I), unbiased expected heterozygosity (uHe), and the number of private alleles (*Npa*) were estimated for each genetic group and subgroups using GenAlEx Version 6.5 [40]. Tajima’s *D* [41] tests and nucleotide diversity (π) were estimated for each genetic group and by chromosome using DnaSP v6 [42]. In general, a Tajima’s *D* value below −2 or above 2 is a strong indication that a locus/gene is not evolving neutrally [43]. The patterns of variation of D and π parameters across the genome within the Mesoamerican, Andean, and Chilean genetic groups were visualized with the ggplot2 package [38].

### 4.5. Linkage Disequilibrium and Haplotype Blocks

Linkage disequilibrium (LD) between each pair of SNPs was estimated using the markers with MAF ≥ 0.05 in the LDcorSV package [44], where the values of *r*^2^ were corrected for population structure and relatedness (*r*^2^_sv_). The kinship matrix was estimated using the rrBLUP package [45]. The LD decay was explained by the nonlinear model proposed by Hill and Weir [46] and adjusted to the NLS function in R software version 4.2.3. Haplotypic blocks were identified using the software Haploview 4.1 [47] based on the confidence interval method described by Gabriel et al. [48] using SNP markers with a MAF ≥ 0.05 and call-rate ≥ 80%.

### 4.6. Candidate Genes Identification and Gene Enrichment

The identification of candidate genes was carried out in the haplotype blocks using the reference genome of *P. vulgaris* (v2.1), arranged in phytozome. Candidate genes were taken using the BioMart tool version 3.14 [49], and information related to gene function, gene ID, and its description was downloaded. Using the gene ID, the FASTA sequence of each gene was obtained. Using Blast2GO v6.0.3 [50], GO terms were obtained for each gene. Blast2GO was also used for GO functional enrichment analysis of the identified genes, describing biological processes, molecular functions, and cellular components.

## 5. Conclusions

This is the first study to characterize the origin of the race Chile of the common bean through its diversity and genetic structure. It was possible to identify a genetically differentiated group that was classified as the race Chile, where the types ‘Coscorrón’, ‘Tórtola’, ‘Sapito’, and ‘Burros’ stand out, which have historically been associated with Chilean beans. It was evident that beans from the race Chile have a close genetic relationship with beans from Argentina, indicating their genetic and geographical origin. In addition, the genetic diversity parameters showed that a reduced population size from Argentina was the precursor of the current race Chile. Then, mutation and selection processes allowed Chilean beans to adapt to local conditions and genetically differentiate themselves from Argentine beans. Various genes were identified that could be associated with the local adaptation of the race Chile. However, more complete studies must be performed for their validation. Our results enabled us to highlight the importance of this unique genetic resource in the world, valuing its nutritional and adaptive properties that are useful in genetic improvement programs considering the requirements of current societies and the effects of climate change, benefiting both the farmers and the consumers. Finally, additional studies using wild race materials from the Andean pool, the evaluation of the genes related to domestication, and comparative genomics can improve understanding the domestication process of the race Chile of common bean.

## Figures and Tables

**Figure 1 ijms-25-04081-f001:**
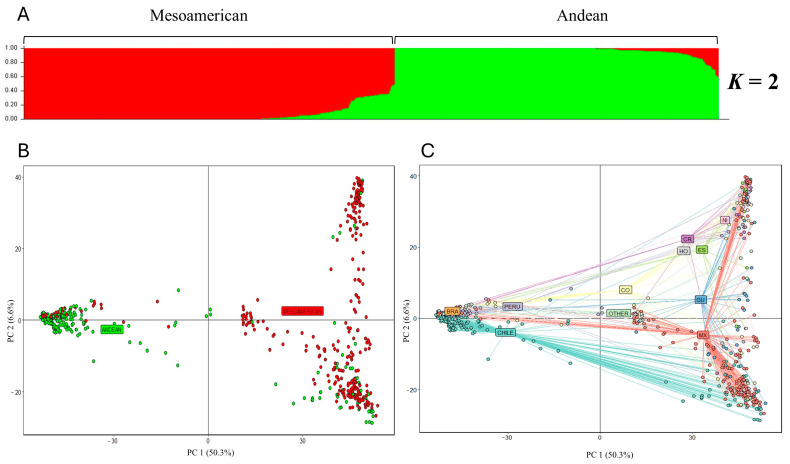
Population structure analysis of the 667 common bean accessions. (**A**) Based on the Bayesian clustering method. The inferred ancestry probability (Q_i_) for each accession is on the Y axis and the accessions is on the X axis. (**B**) Principal component analysis based on gene pool. The first two principal components (PCs) of analysis accounting for 50.3 and 6.6% of total variation, respectively. Green and red accessions correspond to the Andean and Mesoamerican gene pool, respectively. (**C**) Principal component analysis of 667 accessions according to their country of origin. CHILE; PERU; BRA: Brazil; CO: Colombia; CR: Costa Rica; HO: Honduras; ES: El Salvador; GU: Guatemala; NI: Nicaragua; MX: Mexico; OTHER: advanced breeding lines from different countries.

**Figure 2 ijms-25-04081-f002:**
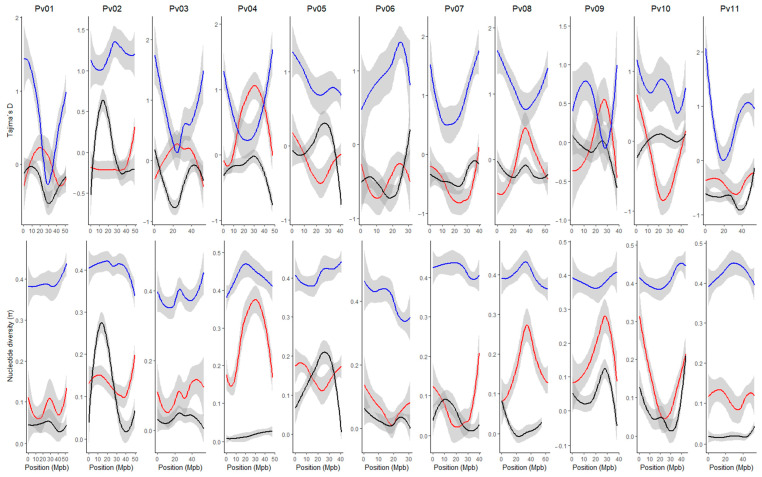
Genome-wide loess curves for nucleotide diversity (π) and Tajima’s D values across all chromosomes in the *P. vulgaris* genome for Mesoamerican (blue), Andean (red), and Chile (black) genetic groups.

**Figure 3 ijms-25-04081-f003:**
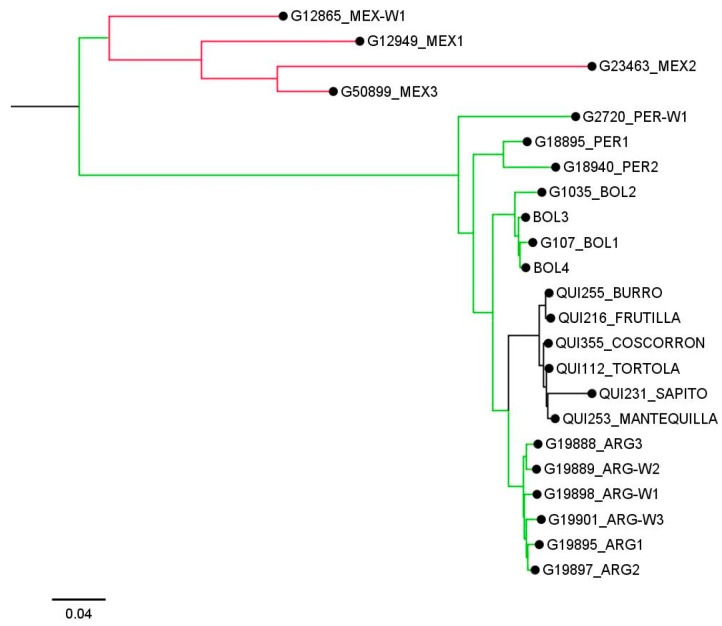
Neighbor-joining tree based on genetic distances between some pure accessions belonging to the race Chile of common bean and others from the Andean genetic pool. Red, green, and black clades indicate accessions belonging to the Mesoamerican, Andean, and the race Chile genetic groups, respectively.

**Figure 4 ijms-25-04081-f004:**
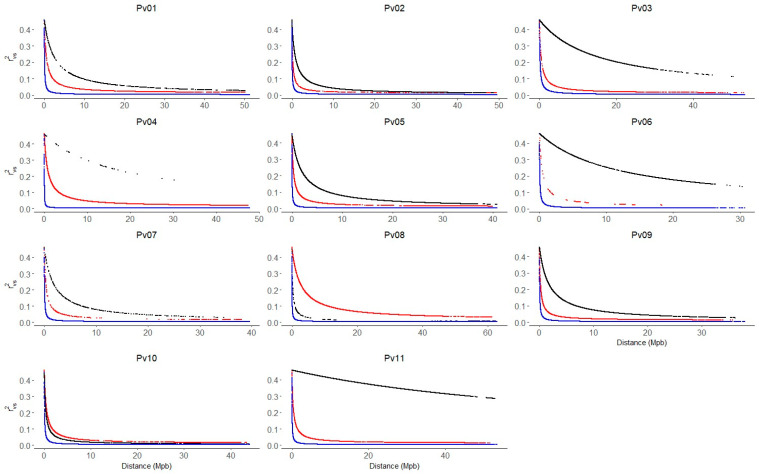
Linkage disequilibrium (LD) decay across the 11 common bean chromosomes (Pv01–Pv11). LD values were corrected by population structure and kinship (r^2^_sv_). The blue, red, and black lines indicate the LD decay for the Mesoamerican (M1 + M2), Andean (A1 + A2) and Chile (A3) genetic group, respectively.

**Figure 5 ijms-25-04081-f005:**
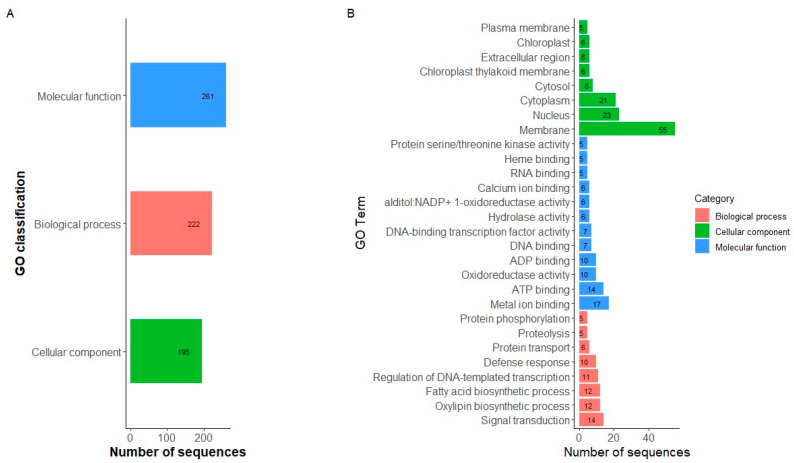
Gene ontology (GO) functional classification. (**A**) GO classification annotated in GO categories biological process, cellular components, and molecular function. (**B**) Histograms of distribution in subcategories of the enriched gene ontology (GO). Only GO subcategories constituted by at least five sequences from the data set in the MF, BP, and CC categories are represented.

**Table 1 ijms-25-04081-t001:** Parameters of genetic and nucleotide diversity for the entire population by gene pool and genetic subgroups identified through structure.

Gene Pool	N	Na	Ne	I	uHe	Npa	D	π	PPL(%)
Andean	310	1.886	1.138	0.177	0.099	-	−0.962	0.097	88.64
A1	10	1.708	1.401	0.366	0.269	342	0.370	0.268	18.52
A2—Peru	86	1.549	1.158	1.165	1.102	61	−0.220	0.102	59.14
A3—Chile	187	1.538	1.073	0.096	0.053	13	−1.371	0.053	87.06
Mesoamerican	350	1.949	1.520	0.446	0.300	-	3.203 *	0.300	94.93
M1	235	1.939	1.485	0.431	0.287	348	2.687 *	0.286	93.87
M2	103	1.687	1.288	0.273	0.177	4	1.157	0.176	83.60
Overall	667	2.000	1.497	0.473	0.308	-	5.504 *	0.403	-

N: number of accessions; Na: Mean number of alleles; Ne: Mean number of effective alleles; I: Shannon´s information index; uHe: unbiased expected heterozygosity; Npa: number of private alleles; D: Tajima’s test of neutrality; π: nucleotide diversity; PPL: percentage of polymorphic loci. * Significant at 5%.

**Table 2 ijms-25-04081-t002:** Identified haplotype blocks in the Mesoamerican (M1 + M2), Andean (A1 + A2), and race Chile (A3) genetic groups.

Chr	Total # of Blocks			Average SNP/Block			Mean Block Size (kb)			Unique Blocks *		
	MESO	AND	Chile	MESO	AND	Chile	MESO	AND	Chile	MESO	AND	Chile
Pv01	20	2	3	3.35	2	4.3	122.2	9.5	228.32	18	1	0
Pv02	29	24	16	2.51	4.7	4.56	58.68	269.37	319.18	4	5	3
Pv03	0	2	0	0	2	0	0	10	0	0	2	0
Pv04	22	19	0	2.13	3.84	0	22	190.42	0	13	12	0
Pv05	20	25	14	3.05	5.04	5.07	73.75	258.48	402.07	5	7	2
Pv06	12	0	0	3.33	0	0	119.75	0	0	12	0	0
Pv07	15	3	2	2.53	2.66	2	38.33	58	6.5	15	1	0
Pv08	20	8	8	2.3	2.5	2.5	53.8	128.5	128.5	19	0	0
Pv09	8	8	1	2.25	4	4	20.12	209.75	495	4	5	1
Pv10	6	11	6	2.33	4.36	7.5	88.16	214.36	361.83	3	6	5
Pv11	20	8	1	2.75	6.12	4	67.25	210	290	12	3	1
Total	172	110	51	2.41	3.38	3.08	60.37	141.67	202.85	105	42	12

* Haplotype blocks that were constructed exclusively on one of the three genetic groups. MESO: Mesoamerican genetic group; AND; Andean group. Pv01–Pv11: represent the 11 chromosomes of *Phaseolus vulgaris* (Pv).

**Table 3 ijms-25-04081-t003:** Genes located within the region (36,195,096–36,365,141 bp) on chromosome Pv05 with a significant Tajima value (D = 2.3).

Gene Name	Description
*Phvul.005G124800*	SERINE/THREONINE-PROTEIN KINASE RIO
*Phvul.005G125000*	AN1-TYPE ZINC FINGER PROTEIN
*Phvul.005G125100*	vacuolar protein-sorting-associated protein 4 (VPS4)
*Phvul.005G125200*	PPR repeat (PPR)//PPR repeat family (PPR_2)//DYW family of nucleic acid deaminases (DYW_deaminase)
*Phvul.005G125300*	protein HIRA/HIR1 (HIRA, HIR1)
*Phvul.005G125400*	FATTY ACYL-COA REDUCTASE 3-RELATED
*Phvul.005G125500*	Long-chain-fatty-acyl-CoA reductase/Acyl-CoA reductase
*Phvul.005G125700*	histidine-containing phosphotransfer protein (AHP)
*Phvul.005G126000*	60S ACIDIC RIBOSOMAL PROTEIN P2-4
*Phvul.005G126100*	PEPTIDYLPROLYL ISOMERASE DOMAIN AND WD REPEAT-CONTAINING PROTEIN 1
*Phvul.005G126300*	AP2 domain (AP2)

## Data Availability

Data is contained within the article and Appendix A.

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
