# Peer review of "A Past Genetic Bottleneck from Argentine Beans and a Selective Sweep Led to the Race Chile of the Common Bean (Phaseolus vulgaris L.)"

_ijms, 2024, doi:10.3390/ijms25074081_

Round 1
Reviewer 1 Report
Comments and Suggestions for Authors
The authors collected 682 common bean accessions and used Illumina 12K Infinium SNP array to identify SNPs, but the authors only used 2,543 SNPs common with the previously published data in the analysis. Other issues include:
Introduction: common bean races should be described in more detail, as well as the information about the five common bean races mentioned in the abstract.
Figure 2 should be integrated with Figure 1.
Line 163: Detailed polymorphic information should be provided in the manuscript.
Figure 5: The colors in the picture require special explanation, and it should be analyzed genomewise.
Author Response
Dear Reviewer;
Please see the attachment

Reviewer 2 Report
Comments and Suggestions for Authors
I congratulate the authors for their work and manuscript. Manuscript ID: ijms-2948514 "A past genetic bottleneck from Argentine beans and a selective sweep led to the race Chile of the common bean (Phaseolus vulgaris L.)" by Arriagada and collaborators describes a deep investigation of the genetic diversity of the common bean. The authors used a vast collection of 667 accessions comprising members of the two main centers of diversity including several countries and thousands of SNP markers. Results are clearly presented and methods are well described including statistical analyses and indices used. The findings can be interesting not only to bean breeders but a wider community interested in GWAS and genomics in general. The common bean race Chile breeding can now be improved based on the knowledge of its evolutionary relationship with other important accessions.
The manuscript is well-written and objective, and perhaps the authors can demonstrate more insights from the genomic comparison data by including an additional figure showing the relationships of the Chilean race to the other main groups and/or accessions. Multiple characteristics could be explored, but perhaps focusing on the region containing the genes of interest could be informative, especially in relation to the types ‘Coscorrón’, ‘Tórtola’, ‘Sapito’, ‘Burros’, and the outer groups from Argentina.
I'll recommend acceptance after minor review solely because of this, to give the authors a final opportunity to include the figure or any other information.
Author Response

(The authors gave the same response as above.)

Round 2
Reviewer 1 Report
Comments and Suggestions for Authors
This manuscript has improved the issues I raised, and the quality of the manuscript has been significantly improved.